# Analysis of Mining-Related Injuries in Chinese Coal Mines and Related Risk Factors: A Statistical Research Study Based on a Meta-Analysis

**DOI:** 10.3390/ijerph192316249

**Published:** 2022-12-05

**Authors:** Jin Tian, Yundou Wang, Shutian Gao

**Affiliations:** Institute of Medical Support Technology, Academy of System Engineering, Academy of Military Sciences, Tianjin 300161, China

**Keywords:** coal mines, injuries, accidents, risk factors, meta-analysis

## Abstract

Background and Objectives: Coal mine injuries commonly occur, affecting both the safety and health of miners, and the normal operation of the coal mine. Accordingly, this study aimed to explore the regularity of injury and injury-related risk factors in coal mines in China so as to establish a scientific basis for reducing the incidence and promoting the prevention and control of injuries. Methods: A meta-analysis of casualty cases and injury-related risk factors from 1956 to 2017 in China was conducted utilizing data from six databases, including CNKI, Web of Science, PubMed, Medline, Embase, and Wanfang data. Summary estimates were obtained using random effects models. Results: There were statistically significant variations in coal mine accident types, types of work, injury sites, age, experience, months, and shifts (*p* < 0.001). Eight types of accidents were susceptible to the risk of injury, and the greatest risk was presented by roof-related accidents (odds ratio (OR) = 0.46, 95% confidence interval (CI) = 0.32–0.6). Coal miners and drillers were at a greater risk of injury (OR = 0.39, 95% CI = 0.35–0.44; OR = 0.22, 95% CI = 0.17–0.26, respectively). The extremities and the soft tissues of the skin were at the greatest risk of injury (OR = 0.44, 95% CI = 0.3–0.58; OR = 0.23, 95% CI = 0.1–0.48, respectively). Compared with other ages, miners aged 21–30 were at a greater risk of injury (21–30 years, OR = 0.45, 95% CI = 0.42–0.47; 31–40 years, OR = 0.29, 95% CI = 0.25–0.32; <20 years, OR = 0.13, 95% CI = 0.03–0.23; >40 years, OR = 0.17, 95% CI = 0.09–0.25). Compared with other miners, those with 6–10 years of experience were at a greater risk of injury (6–10 years, OR = 0.29, 95% CI = 0.25–0.32; 2–5 years, OR = 0.33, 95% CI = 0.25–0.41; <1 year, OR = 0.22, 95% CI = 0.08–0.33; >11 years, OR = 0.22, 95% CI = 0.17–0.27). During the months of July to September, the risk of injury was elevated (7–9th months, OR = 0.32, 95% CI = 0.25–0.39; 10–12th months, OR = 0.24, 95% CI = 0.16–0.31; 1st–3rd months, OR = 0.22, 95% CI = 0.16–0.28; 4–6th months, OR = 0.21, 95% CI = 0.16–0.27). In the three-shift work system, the risk of injury was higher during night shifts (22:00–06:00, OR = 0.43, 95% CI = 0.3–0.56; 14:00–22:00, OR = 0.3, 95% CI = 0.23–0.27; 06:00–14:00, OR = 0.27, 95% CI = 0.18–0.35). Conclusions: The results of this research study reveal that coal mine injuries are prevalent among coal miners. These injuries are often related to the age, experience, months of work, and the three-shift work system of miners.

## 1. Introduction

Chemical, physical, and ergonomic hazards are pervasive in the mining industry, frequently increasing the risk of injury among miners [1]. Recent statistics show that the coal mining industry workforce is burdened with occupational injuries [2]. According to International Labour Organization estimates, mining employs around 1% of the global workforce but accounts for 8% of the global work-related fatalities [3,4]. China’s Seventh National Economic Census revealed that 3.68 million people worked in the Chinese coal industry in 2019 [5]. Major global coal producers face fatal accidents. Specifically, the highest number of fatalities in the coal mining industry in 2017 was related to China (375), followed by India (61), the USA (15), the Republic of South Africa (10), and Australia (3) [6].

An accident at work or occupational accident is an event that occurs while working that may result in physical or mental harm such as injury, illness, or death [7]. For example, on 13 May 2003, 86 and 28 miners were killed and injured, respectively, when a gas explosion occurred in Luling Coal Mine in China. On 28 March 2010, 153 miners were trapped underground after a water gushing accident in a Wangjialing coal mine. Mine injuries have a major impact on workers’ lives and family well-being, raising a range of social issues. Thus, the lives and health of miners deserve widespread attention.

Although prevention mechanisms have been proposed and established, questions arise regarding why injuries continue to occur, why accidents have not been stopped, and what are the risk factors that contribute to injuries in Chinese coal mines. These questions deserve to be further discussed.

Nurbek et al. [8] used statistical analyses to analyze occupational injuries and diseases in the mining industry in the Republic of Kazakhstan between 2008 and 2018, noting that good working conditions, attention to employee welfare, and the improvement of monitoring indicators related to labor protection contributed to a reduction in the level of industrial injuries and occupational diseases. Erdal Özer et al. [9] employed statistical analyses to assess fatal accidents at Zonguldak Coal Mine in Turkey during 2005–2008, assessing the age of the cases, the mechanisms of injury, the wound sites, the causes of death, and the legality of the mine. They concluded that such an assessment could be relevant in reducing the accident rate. Chipo Chimamise et al. [10] conducted a cross-sectional study to analyze factors associated with serious injuries in mines in the Republic of Zimbabwe in 2010, indicating that factors such as worker fatigue and inattention increased the risk of serious injuries. Additionally, Thirumala et al. [11] studied the effects of worker stress on worker performance and injuries, highlighting that sleep-deprived individuals were more likely to suffer injuries, especially bruises and puncture wounds. In another study by Friedman et al. [12], it was found that long working hours and fatigue could increase work-related injuries.

There are many examples in the nuclear field where ”external” errors or omissions have been significant contributors to accidents [12,13]. The decisions and actions of those in the ”complex system” are moderated by the decisions and actions of those outside of that system [13]. The system of coal production is a complex social–technological system with a dynamic equilibrium [14]. Generally, the presence of any negative effect disrupts the equilibrium and results in casualties. The recurrence of mining-related injuries, however, cannot be attributed to a single factor [8,9]. Coupling and correlation with existing factors in a multidimensional space under suitable conditions and time, accidents and injuries will occur [15,16].

In mining, the agent factors are the hazards, while the host factors include individual characteristics (personal factors) and behavioral factors. Additionally, environmental factors and job-related risk factors (organizational factors) are involved [17,18,19].

Considering the complexity and diversity of the influencing factors associated with casualties, a great deal of research has been conducted on the identification of human factors. The human factors of workers are primarily divided into three categories: individual characteristics, mental state, and physiological state [15]. General individual characteristics consist of gender [20], age [21], lack of experience [22,23], and educational level [10,24]. In addition, the unsafe behavior of miners is closely related to the mental state, safety consciousness [25], safety attitude [15], cognitive biases [15,26,27], mental stress [28], and other factors. Physiological factors, such as insomnia and fatigue, have also been artificially associated with personal insecurity [29].

It has been reported that an unfavorable work environment and poor working conditions are occupational injury risk factors [30]. Physical, chemical, and biological contamination may represent an environmental hazard [8]. The physical and mental health of miners is gravely endangered by hazardous working conditions. Additionally, longer work hours are associated with injuries. The risk of occupational injury was 15% lower during night shifts than during morning shifts according to a study that evaluated the relationship between work hours and injury risk. The risk of occupational injury was 15% higher in 10 h workdays than in 8 h workdays. On the other hand, 12 h workdays increased the risk of occupational injury by 38%. When working more than 12 h a day, occupational injuries increased by 147% according to a previous study [31].

There is a correlation between the level of management or supervision and poor working conditions [18]. Individual and organizational factors were discovered to be tightly connected and intertwined in the dynamic mechanism influencing employee behavior [32]. For instance, in artisanal and small-scale mining operations with limited management or government supervision, the sites were rife with hazards, and the risk of injury was higher than that in large-scale mining operations [33]. As a major component of organizational factors, environmental support, including colleagues’ concern, leaders’ commitment, and a safety climate, has garnered the attention of many scholars because it may be the leading cause of safety accidents [34].

Over the past decades, there has been a growing emphasis on the significance of safety culture in preventing occupational injury and disease [35]. Some reports suggested that cultural factors (rather than just management systems, policies, and procedures, or more technical aspects of safety) can play a significant role in explaining the frequency and severity of injuries and diseases within an organization [36]. A strong, healthy safety culture provides an environment in which the probability of this kind of failure is significantly reduced. However, it is not eliminated, due to the inherent sensitivity of any organizational culture to senior management influences [37]. Safety culture is the degree to which managers and decision makers prioritize safety. Studies have demonstrated that safety culture plays a vital role in organizational safety management and has a substantial impact on the success or failure of security management. In general, it is believed that safety culture influences the attitudes and behaviors of employees in relation to an organization’s continuous health and safety performance. In the coal mining business, safety culture has become a major focal point [38].

There is a causal relationship between corporate culture and organizational performance, and culture can influence organizational performance under specific conditions. Additionally, cognitive and motivational biases play a significant role in organizational performance [39]. Some studies have shown that organizational factors play a key role in creating conditions for triggering major accidents (aviation, railway transportation, nuclear industry, oil exploitation, mining, etc.). Komljenovic Dragan et al. showed that the role of organizational performance appears as a determining factor in creating unfavorable conditions leading to a “drift to failure” through eroding safety margins throughout organizations [40].

Many recent event analyses across various industries have shown that organizational factors play a key role in creating conditions for triggering major accidents [40]. A real reduction in accident frequency requires getting to the bottom of human and organizational performance issues [40]. Some studies have suggested that there might be situations in which human performance, in relation to adhering to laws, principles, and theorems of systems theory, is affected under three conditions: (1) knowing systems theory but choosing to ignore it, (2) knowing systems theory but having poor execution, and (3) not knowing systems theory [41].

Research has revealed that perceived motivational biases and cognitive biases influence the judgment and decisions of both laypeople and experts [42]. Motivational bias is described as the influence of the desirability or undesirability of an event, consequence, outcome, or choice on assessments [43,44]. In the process of management and safety risk decision making in coal mine organizations, decision makers are influenced by insufficient experience, inadequate implementation of responsibilities, and limited life value assessment and deliberately make optimistic predictions about preferred actions or results, resulting in poor outcomes; it is difficult to correct this motivational bias. Cognitive bias is a systematic discrepancy between the “correct” answer to a judgmental task, given by a formal normative rule, and the decisionmaker’s or expert’s actual answer [45]. Analyzing a vast number of historical cases reveals that cognitive biases are prevalent in the course of accidents. In coal mine safety production systems, humans making unwise decisions in uncertain situations due to their limited understanding can result in catastrophic accidents. Four specific manifestations of cognitive bias have been identified: representation bias, availability bias, overconfidence, and attribution bias [46]. Currently, the majority of research focuses on the outcome-driven bias [42]. Cognitive bias can, in fact, help us to judge the results of accidents and predict their causes, on one hand, and help us to explain certain phenomena and explain the reasons according to the phenomena, on the other hand. The study of the motivational bias and the cognitive bias provides new ideas and methods for coal mine safety management.

Occupational injuries are common in the coal mining industry, and miners have some understanding of the dangers of their work. Nonetheless, knowing dangers does not mean that the person can prevent and eliminate them. Thus, understanding the characteristics of coal mine injuries and risk factors is essential for developing policies to protect against technological disasters and improve injury prevention. On one hand, the statistical analysis of coal miners’ injuries in China can provide a factual basis and methodological guidance for subsequent prevention, disaster prevention and control, accident rescue, and treatment of personnel. On the other hand, epidemiological research methods have been widely applied to various areas of human-related injuries, including earthquake disasters and road traffic.

A systematic and comprehensive analysis of the health and safety of mine workers was performed by applying the existing and proven risk analysis tools for epidemiological surveys in relevant fields in this study. The analysis of the epidemiological risk factors for mine workers can be used to provide advice and recommendations for the development of mine management, and laws and regulations in China.

This study sought to investigate the characteristics of injuries in China’s coal mines, mainly including accidents and types of work injuries, as well as to analyze the risk factors associated with coal mine injuries (miner age, experience, shifts, and months). The risk factors for this study were selected based on their importance in previous studies. After presenting/describing the first part, the existing parts are organized as detailed below.

The second part mainly discusses the method of research application (meta-analysis), data source screening, and literature quality assessment. In addition, the third part describes the effects of coal mine accident types, injury types, injury sites, and some injury-related risk factors. The relationship between the characteristics of China’s coal mine injuries and risk factors are mainly discussed in part four. Finally, part five summarizes the main content of this paper and looks forward to future research content.

## 2. Methods

A meta-analysis is a statistical technique that combines the results of different studies on the same topic [47], that is, a quantitative, scientific synthesis of research findings [48]. The advantage of this method is that it increases the statistical strength and precision of the estimated effects by combining the results of previous studies, overcoming the problems of small sample sizes and insufficient statistical strength [48]. Meta-analyses have a wide range of applications, especially in epidemiological characterization; this method was employed in this study.

### 2.1. Data Sources and Search Strategy

Published observational studies on the casualties among Chinese miners were included in the meta-analysis. A systematic search of the literature published between January 1965 and April 2021 was performed using different databases, such as CNKI, Wanfang, PubMed, Web of Science, Embase, and Medline. Medical subject headings and keywords used as search terms were “coal mine” or “miners”, in combination with “trauma”, “wound”, “hurt”, “multiple injury”, “compound injury”, “casualty”, “occupational disease”, or “industrial injury” in the title, abstract, or keywords. Additionally, traced references were included in the literature to access the relevant literature. We attempted to contact the authors via email if some of the statistics in the literature were incomplete. Every included article was also hand-searched to make sure that no additional studies were missed during the period of interest.

### 2.2. Eligibility Criteria

Several criteria were considered for the eligibility of articles for inclusion in the meta-analysis. The study subjects had to be Chinese coal miners, including both men and women. Furthermore, the injuries of miners had to be caused by an underground mine instead of an open-pit mine. Moreover, the statistical scope was acute rather than cumulative injuries. Furthermore, the articles had to contain information on injury characteristics and the causes of injuries and had to analyze the risk factors associated with coal mine casualties in China. Finally, the literature had to deal with primary data. On the other hand, the exclusion criteria included investigations of pneumoconiosis in miners (pneumoconiosis [49] is a disease caused by the inhaling of coal dust, rock dust, and other dust for a long period of time, causing fibrosis of the lung tissues in workers; thus, this disease is not caused by accidents). In addition, studies on mental disorders among miners due to coal mine casualties and articles focusing on the coal mine casualties associated with other countries and including data with low reliability or poor quality, as well as review articles and reports, were among the excluded studies.

### 2.3. Research Selection

Figure 1 depicts the complete four-step process for the selection of the articles. In the first stage, the database was searched using “search terms”, and 1828 records were retrieved accordingly. However, 1618 articles were excluded after reading the titles and abstracts of the articles because there were no specific classification data of coal mine injuries in China. Accordingly, 201 articles entered the full-text content selection stage, and their full texts were read by the researchers. This was because most of these articles did not specifically examine the human risk factors associated with mining-related injuries. Subsequently, 175 articles were excluded, and 35 articles were included in the analysis.

### 2.4. Data Extraction and Quality Assessment

Relevant information from 36 articles was extracted by the lead author using a data extraction Microsoft Excel spreadsheet template. The template was developed in accordance with the Strengthening the Reporting of Observational Studies in Epidemiology (STROBE) guidelines [50] to maintain consistency and uniformity in the review [51]. The STROBE statement contains 22 items designed to extract data from observational studies [50]. The information on authors, publication date, study method, sample size, study variables, evaluation analysis, etc., was obtained from the 35 articles. The nature of coal mining enterprises included state-owned and privately owned coal mines.

The quality of the 35 articles was assessed according to the Newcastle–Ottawa Scale (NOS), which was explicitly designed to assess the quality of non-randomized studies. NOS content validity and inter-rater reliability were confirmed in previous research [52]. In addition, the NOS establishes predefined criteria for case-control and cohort studies, but it has been further developed to include cross-sectional and longitudinal studies [53,54]. The quality of cross-sectional and longitudinal studies was evaluated using the NOS criteria in Table 1.

### 2.5. Data Analysis

Random effects models were considered to derive pooled effect sizes (ORs) in the meta-analysis using Stata, version 12.1. This model was employed since it assumes a difference in effect sizes among studies but leads to an apparently conservative null hypothesis model [89] and takes into account the impact of specific objects [90,91,92]. Statistical values were pooled in the meta-analysis for the type of coal mine accident, type of work causing injury, site of injury, and human risk factors. Pooled combined effect sizes (ORs) and 95% confidence intervals (CIs) were calculated, and the heterogeneity among studies was assessed using the Cochran Q test and the I^2^ statistical test [93]. In all analyses, *p* < 0.001 was considered to be statistically significant. The injured part was identified primarily using its physical characteristics. In the current analysis, four levels of the variable “age” were considered:. (<20), (21–30), (31–40), and (>40). The variable “the miner’s work experience” similarly included four levels: (0–2), (2–5), (6–11), and (>11). The variable “shifts” included three levels: (06:00–14:00), (14:00–22:00), and (22:00–06:00).

## 3. Results

Figure 2 illustrates the results of the identification, screening, eligibility, and inclusion processes. Despite a large number of searched articles, less than 1% met the inclusion criteria, including 32 articles (*n* = 31 Chinese and *n* = 1 English published in 1984–1999 (11), 2000–2009(11), and 2010–2018(11)) on coal mine casualties in China and 3 articles on human risk factors related to coal mine casualties in China.

Table 1 presents the methodological quality assessment of the study. These studies investigated the characteristics of coal mine accidents, the type of work causing the injury, the location of the casualty, and the relationship between the casualties and the risk factors. Based on the data in Table 1, the quality of the included articles was acceptable, with a maximum of seven stars.

There were statistically significant differences in different types of coal mine accidents, injury types, injury sites, age, experience, and shifts (*p* < 0.001).

### 3.1. Main Types of Coal Mine Accidents

Eight types of accidents had a higher probability of occurring in China; the highest risk was that of roof-related accidents (OR = 0.46, 95% CI = 0.32–0.6), fire-related accidents (OR = 0.23, 95% CI = 0.14–0.23), and coal and gas explosions (OR = 0.125, 95% CI = 0.9–0.16), respectively. The other types were electromechanical (OR = 0.09, 95% CI = 0.07–0.11), blasting-related (OR = 0.07, 95% CI = 0.03–0.09), transportation-related (OR = 0.06, 95% CI = 0.02–0.1), flooding-related (OR = 0.03, 95% CI = 0.01–0.04), and other (OR = 0.16, 95% CI = 0.13–0.18) accidents, respectively.

### 3.2. Types of Work

Coal miners and drillers (OR = 0.39, 95% CI: 0.35–0.44; OR = 0.22, 95% CI: 0.17–0.26) were at the highest risk of injury, followed by mine transporters (OR = 0.17, 95% CI = 0.14–0.21), and auxiliary (OR = 0.13, 95% CI = 0.1–0.16), electromechanical (OR = 0.05, 95% CI = 0.04–0.07), and other types of workers (OR = 0.19, 95% CI = 0.15–0.23), respectively. Other types of workers included coal preparation workers, casual repairmen, carriers, and blasters.

### 3.3. Injured Parts of the Body

There are various classifications for the location of injuries based on Classification of Casualty Accidents of Enterprise Workers (GB6441-86); the considered body parts were the head, spine, limbs, torso, chest, abdomen, skin, and soft tissue. Extremities were the most vulnerable body parts (OR = 0.44, 95% CI = 0.3–0.58), including soft tissues of the skin (OR = 0.23, 95% CI = 0.01–0.48), head (OR = 0.18, 95% CI = 0.15–0.2), trunk (OR = 0.14, 95% CI = 0.09–0.19), chest (OR = 0.1, 95% CI = 0.08–0.11), spine (OR = 0.09, 95% CI = 0.07–0.11), and abdomen (OR = 0.07, 95% CI = 0.06–0.09), respectively.

### 3.4. Risk Factors Associated with Casualties

Severe characteristics, such as the personal features of miners, have been reported and investigated in previous studies. However, two personal factors (age and experience), which emerged from the articles, associated with casualties were reviewed; these factors are also often discussed in the mining industry.

There was a relationship between the age of miners and injury risk. The risk of injury was higher in miners aged 21–30 (OR = 0.45, 95% CI = 0.42–0.47) and 31–40 (OR = 0.29, 95% CI = 0.25–0.32) years than in those aged 30–39 years; the risk of injury was low in the other age groups (<20 years, OR = 0.13, 95% CI = 0.03–0.23; >40 years, OR = 0.17, 95% CI = 0.09–0.25). The risk of injuries decreased with age.

Likewise, a relationship was found between the experience of miners and injury risk. The risk of injury was higher in miners with 6–10 (OR = 0.29, 95% CI = 0.25–0.32) and 2–5 (OR = 0.33, 95% CI = 0.25–0.41) years of experience than miners with less than 1year or more than 11years of experience (<1 year, OR = 0.21, 95% CI = 0.08–0.33; >11 years, OR = 0.22, 95% CI = 0.17–0.27).

Severe characteristics, including job-related factors in miners, have been reported and evaluated in previous research. However, two factors (shifts and months), which emerged from the articles, related to casualties were reviewed; these factors are also frequently discussed in the mining industry.

Based on the findings, a relationship was observed between the shifts and the injury risk. The risk of injury was higher in the shift between 22:00 and 06:00 (OR = 0.43, 95% CI = 0.3–0.56), while the other two shifts (14:00–22:00 and 06:00–14:00) showed a low risk (OR = 0.3, 95% CI: 0.23–0.27; OR = 0.27, 95% CI: 0.18–0.35).

The risk of injury was higher between the 7th and 9th (OR = 0.32, 95% CI = 0.25–0.39), and 10th and 12th (OR = 0.23, 95% CI = 0.16–0.31) months, whereas the other months presented a lower risk of injury (1–3, OR = 0.22, 95% CI = 0.16–0.28; 4–6, OR = 0.21, 95% CI = 0.16–0.27; Figure 2). In terms of injury time, the outcome was always consistent with the expectations.

## 4. Discussion

Coal mine injuries have attracted the attention of many experts in the world. Ngoy Kalenga [94] investigated trends in injury rates and causes of death in the Japanese mining industry, reporting median injury, serious injury, and fatality rates of 129.29, 5.44, and 2.99 per 1000 workers, respectively. Although coal mine injuries have been studied in many countries, the complex geology, the deep burial of coal resources, the predominance of shaft mining in China’s coal mines, and the difficulty of mining make the types of injuries caused by accidents different from those abroad. Injuries result from complex and diverse factors. In order to improve mine safety, it is necessary to further discuss the relationship between injuries and influencing factors.

### 4.1. Accident Types: Casualties

Accidents cause injuries, and the statistical analysis of accident types is the basis for hazard identification and injury prevention. It can also determine which accidents are most likely to result in casualties. A higher frequency of accidents leads to more casualties and harm. Based on the above-mentioned analysis, the most common types of accidents in China were found to include roof-related, fire-related, coal and gas protrusion, gas explosion, and mine water-related accidents, which is in line with the findings of previous studies [95,96,97]. This indicates that the task of disaster prevention and mitigation in coal mines is extremely arduous, and it is necessary to pay special attention and prevent the occurrence of these accidents.

### 4.2. Types of Work: Casualties

The contributing factors to injuries during work is the focus of the discussion The question of which jobs are the main sources of injury in coal mines needs to be answered. The analysis of injury types caused by coal mining shows that coal mining personnel report a higher number of injuries. The reason could be the development of mechanization in mining process Before 1986, the technology of general mining was outdated, and coal miners and drillers were vulnerable to being smashed by falling roofs and due to rib spalling, with a high probability of injury. The type of coal mine injury is also determined by the distinction between mechanized and conventional coal mining techniques.In 1986, the degree of automation and mechanization of coal mining increased with the implementation of full mechanization in mining, and the rate of casualties showed a significant reduction. It can be guessed that improving the degree of coal mining mechanization is an important condition for reducing casualties.

### 4.3. Location of the Injury

The risk of injury to the extremities was found to be the highest, followed by that of injury to the head. This finding conforms to those of previous studies; for example, hands and fingers were the most commonly injured body parts in miners in Zambian coal mines [98]. Regarding the fingers, some narrative observations demonstrated that the thumb, index, and middle fingers were the most injured parts [99]. The head and torso are also vulnerable to injury. Likewise, Laflamme and Blank [100] evaluated injuries in Swedish underground mines during 1980–1993 and found that wrists, hands, and fingers were the most affected body parts (28%). Moreover, Faisal M. Alessa [101] assessed the number and severity of wrist, hand, and finger injuries among miners in the U.S. mining industry from 2000 to 2017, reporting a decrease in total hand injuries and a significant correlation among fracture, amputation events, and serious injuries. Several academic [101] studies have demonstrated that fracture injuries are considered to be among the most common hand injuries in miners and various other occupational settings.

We speculate that the damage caused by external forces is related to the working posture of the worker. Standing is a common position during operation, and miners’ limbs have a large range of motion. The materials or tools are bulky and hard, and may touch, hit, crush, or press against the body, causing injury.

Injuries can be controlled and prevented with suitable physical protection and organizational management. An important means of protecting miners against contact injuries (such as hand bumps, head injuries, etc.) is the use of suitable personal protective equipment. The proper use of PPE (personal protective equipment) can avoid 37.6% of occupational injuries and diseases, according to OSHA (National Institute of Occupational Safety and Health). In total, 12–14% of occupational injuries resulting in total disability were reported to be caused by employees not wearing the appropriate PPE [102]. Miners are advised to wear the appropriate PPE to protect themselves against physical impact, mechanical strikes, or falling objects. Effective safety leadership, safety atmosphere management, and cultural systems are effective measures for intervening in unsafe worker behaviors [103]. Critical to achieving a positive outcome are the safety training, support, goal setting, feedback to stimulate the application of newly acquired knowledge, and incentives or rewards to reinforce performance provided by management [104,105]. The realization of mechanization and intelligence is an essential method for reducing injuries. With the development of information technology, it is recommended to incorporate intelligent technology into security management; for example, wearable sensors can monitor workers’ individual behaviors [106] and an adaptive heuristic mathematical model based on the IoT (Internet of Things) and RFID (Radio Frequency Identification) can be used as a real-time monitoring system.

### 4.4. Shift-Related Injuries

The working mode of “three shifts” in China’s coal mines began in 1949, and miners need to take turns to cover the morning, mid-day, and night shifts (three shifts) within 24 h. The results revealed that the night shift presented a higher risk of casualties, followed by the middle and morning shifts. The conclusion that the night shift (22:00–06:00) has a higher probability of casualties is different from that of previous studies. Specifically, Zhang Zhiqiang [107] indicated that 09:00–13:00 was the peak period of mine accidents. Most injuries in Ghana’s underground mines were reported to happen during the morning shift [44]. Similarly, most injuries in Serbian mines occurred in the morning shift (07:00–15:00) [108]. Eric Stemn [109] used descriptive statistics to analyze the data of 202 reported injuries in the mining industry in Ghana over the past 10 years, indicating that approximately 75% of injuries occurred during early-shift hours and that 86% of injuries and 90% of deaths were related to mining equipment and machinery.

It is estimated that the definition of shift time and data statistics affect the difference in conclusions. On one hand, different countries have a variety of time divisions for three shifts. On the other hand, the sample size of casualties related to the shifts may not be large enough to exclude the interference of other factors

Overall, regardless of which shift represents the highest risk of casualties, the above does at least suggest that there is a relationship between shift work systems and casualties. The shift system not only affects the physical and mental health of miners but also reduces their productivity and personal well-being [110,111]. A lack of adequate or regular sleep influences alertness, reaction time, hand–eye coordination, and other cognitive abilities in shift miners, leading to fatigue, reduced work productivity, unsafe behavior, and accidents [112,113]. Night-shift work in a coal mine typically entails a prolonged period of cognitive activity, which can cause miners to experience heavy mental fatigue [114]. It reduces cognitive performance and makes individuals more readily distracted during psychomotor vigilance tasks [115]. It is recommended that miners get sufficient rest and recover their energy after night shifts, that mine teams establish a more reasonable shift work system, that mine managers optimize the quality of logistics services, that miners’ food quality and standards be improved, and that nutritional supplements be strengthened.

In recent years, with the in-depth advancement of overcapacity work and the release of high-quality production capacity in China, many mines do not need “three shifts” to ensure production. A new model of “cancelling night shifts + Sunday rest” has been implemented in some mines; for example, Zaozhuang Mining Group cancelled night shift operations in 11 mines. These measures could help to reduce coal mine casualties.

### 4.5. Month-Related Injuries

In China, the risk of injury was found to be higher from July to September than in the other months. This article considers several presuppositions, such as crop harvest from July to September, with rural coal miners living in the countryside being busy with farm work and physical fatigue affecting their concentration and increasing the risk of accidents. Unfortunately, this study lacks field investigations to verify this presupposition, which will be the focus of the next study. For different months representing various risks of injuries, the manager of the coal mine needs to develop a scientific arrangement of work, particularly by paying attention to arranging work reasonably during the frequency period. Furthermore, miners should be given psychological counselling to ensure that they work with negative emotion and maintain the optimal working status.

### 4.6. Age- and Experience-Related Injuries

In China, miners of different ages were found to be at risk of casualties, especially those in the age range of 21–30 years. In response to this phenomenon, coal mining enterprises should provide safety training for workers of all ages to improve their safety awareness while reducing casualties.

The risk of injuries in workers with 6–10 years of experience was found to be twice as high as that in workers with less than 1 year of experience, implying that extensive work experience is not inversely proportional to injury. Previous studies have reported that elderly employees are at a higher risk of suffering severe injuries [116]. Kecojevic et al. [117] concluded that miners with less than five years of experience were involved in 44% of accidents. It was supposed that miners with long working years rely more on work experience and obey rules and regulations far less than young miners, resulting in unsafe behaviors. It is advised that miners maintain a calm and cautious attitude at work. Meanwhile, regular vocational training can assist inexperienced workers in becoming familiar with and knowledgeable about their tasks, thus lowering injuries and deaths.

Why do miners of different ages and with different levels of experience sustain different injuries? For coal mine casualties, the analysis results are not as beneficial as an evaluation of the process with an inference mindset, which could aid safety management and prevent injuries at the root. This issue can also be considered from the perspective of the cognitive biases. The first is the representative bias, whereby miners compare their own or others’ operational experience and give it a greater weight in decision making, view experience as science, and treat small changes as normal. The second is availability bias, which will induce miners dealing with tasks negatively. It manifests as a fluke of laziness in miners’ operation and entices miners to complete tasks by experience and luck rather than by rules Another aspect is overconfidence, which is due to the fact that miners believe they have higher working skills and are able to handle various risks they encountered and are optimistic about taking risks as if they were normal. Moreover, the attribution bias refers to miners unintentionally or completely deliberately attributing to accidents In fact, miners are affected by various factors during on-site operations, and studying the relationship between coal mine injuries and influencing factors from the perspective of the cognitive biases is helpful for safety management.

## 5. Conclusions

Numerous studies have demonstrated that coal mine injuries are influenced by a variety of factors. The meta-analysis method was used to retrospectively analyze the characteristics of coal mine accident types, types of work, and the location of injuries, along with some human risk factors related to injuries, in the Chinese context. Eight types of accidents were demonstrated to present a high risk of casualties in China, especially roof- and fire-related accidents. Based on the findings, coal miners and diggers were at found to be at the highest risk of injury. Limbs were determined to be at a higher risk of injury than the other parts of the body. Although the types of injuries and injury sites associated with casualties will likely change as coal mine mechanized production technology advances, enhancing the safety and intelligence of equipment and optimal working environment conditions are crucial initiatives for reducing casualties. Factors such as age, experience of miners, “three-shift” work system, and months were found to be associated with the occurrence of coal mine injuries. Younger miners were found to be at a greater risk of injury than older miners, and it is recommended to improve miners’ awareness by enhancing safety management and training.

Interventions that avoid these factors may be beneficial. This study had some limitations. In future research, it is suggested that the omitted factors, such as educational background, marital status, family conditions, and the cognitive biases of miners, are included in the discussion; furthermore, the analysis model should be improved to determine whether these factors have a significant impact on the injury of Chinese mine workers from qualitative and quantitative perspectives.

## Figures and Tables

**Figure 1 ijerph-19-16249-f001:**
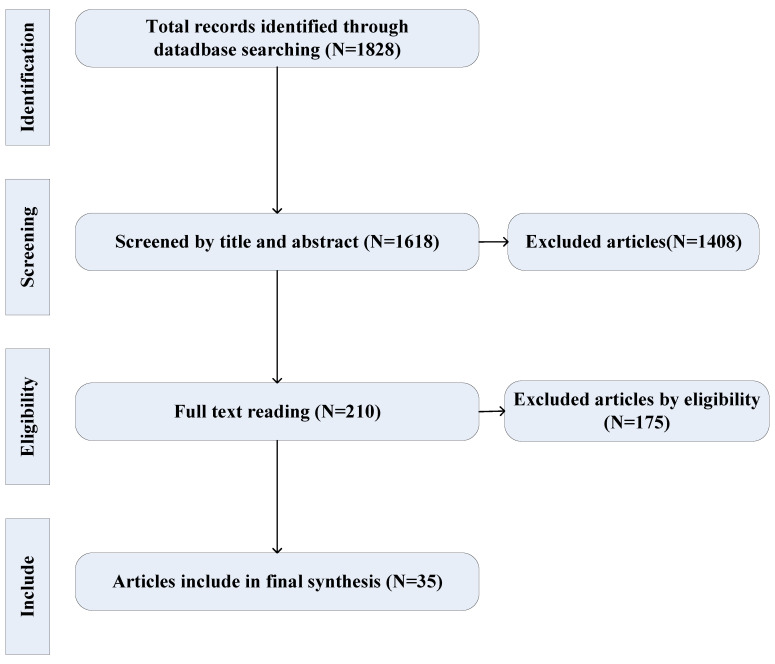
Flowchart for selecting articles on injuries and risk factors associated with coal mine casualties in China.

**Figure 2 ijerph-19-16249-f002:**
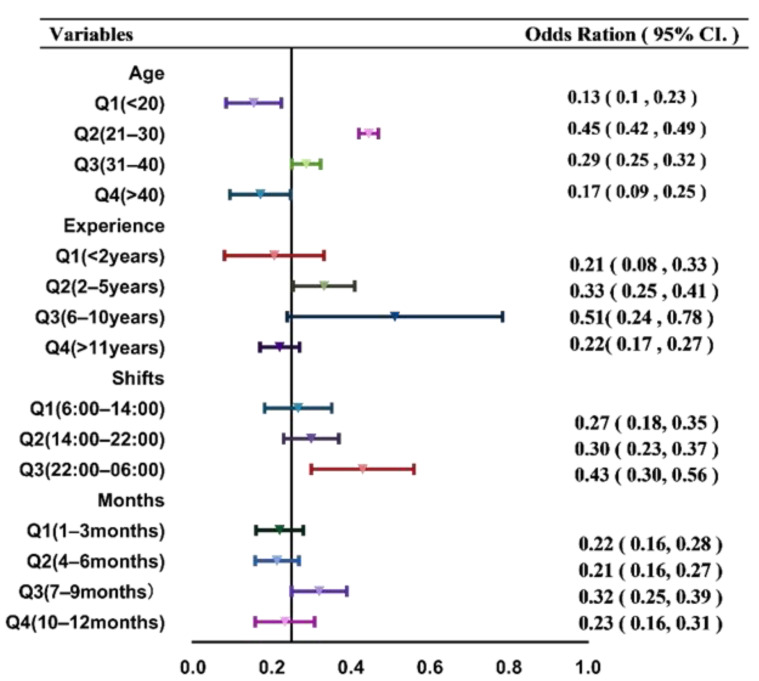
Associations of injuries with several risk factors among coal miners in China.

**Table 1 ijerph-19-16249-t001:** Results of the critical appraisal of cross-sectional and longitudinal studies.

	Literatures	Selection				Comparability	Outcome		
		Representativeness of Sample	Sample Size	Non-Respondents	Ascertainment of Exposure	Based on Design and Analysis	Assessment of Outcome	Statistical Test	Overall Quality Score
1	[55]			*	*	*	*	*	5/7
2	[56]	*	*	*	*		*	*	6/7
3	[57]	*	*	*	*		*	*	6/7
4	[58]	*	*	*	*		*	*	6/7
5	[59]	*	*	*	*		*	*	6/7
6	[60]	*	*	*	*		*	*	6/7
7	[61]			*	*		*	*	4/7
8	[62]		*	*	*		*	*	5/7
9	[63]	*	*	*	*		*	*	6/7
10	[64]		*	*	*	*	*	*	6/7
11	[65]	*	*	*	*		*	*	6/7
12	[66]		*	*	*	*	*	*	6/7
13	[67]	*	*	*	*		*	*	6/7
14	[68]		*	*	*		*	*	5/7
15	[69]	*	*	*	*	*	*	*	1
16	[70]	*	*	*	*		*	*	6/7
17	[71]		*	*	*	*	*	*	6/7
18	[72]		*	*	*		*	*	5/7
19	[73]		*	*	*	*	*	*	6/7
20	[74]		*	*	*	*	*	*	6/7
21	[75]	*	*	*	*		*	*	6/7
22	[76]		*	*	*	*	*	*	6/7
23	[77]	*	*	*	*		*	*	6/7
24	[78]	*		*	*		*	*	5/7
25	[79]	*	*	*	*	*	*	*	1
26	[80]		*	*	*	*	*	*	6/7
27	[81]		*	*	*		*	*	5/7
28	[82]	*	*	*	*		*	*	6/7
29	[83]		*	*	*	*	*	*	6/7
30	[84]	*	*	*	*		*	*	6/7
31	[85]	*	*	*	*	*	*	*	1
32	[86]		*	*	*		*	*	5/7
33	[20]		*	*	*	*	*	*	6/7
34	[87]		*	*	*	*	*	*	6/7
35	[88]		*	*	*	*	*	*	6/7

Definition of set criteria [15]: 1. Representativeness of the sample (i.e., truly representative of the average of the target population—all subjects or random sampling)—star (*) awarded. 2. Sample size (i.e., justified and satisfactory)—star awarded. 3. Response rate was assessed and was satisfactory (>70%)—star awarded. 4. Ascertainment of exposure (risk factor), i.e., validated measurement tool or non-validated measurement tool but tool available or described—star awarded. 5. Comparability, i.e., the study controlled for potential confounders, or subjects in different groups were compared based on design or analysis—star awarded. 6. Outcome, i.e., independent blind assessment, record linkage self-report, or no description—star awarded. 7. Statistical analysis, i.e., statistical test used to analyze the data was correctly described and appropriate, and the measure of association was presented, including confidence intervals and *p*-values—star awarded.

## Data Availability

This is the research based on published literatures. The data are collected in these literatures. The data presented in this study are available in references [20,55,56,57,58,59,60,61,62,63,64,65,66,67,68,69,70,71,72,73,74,75,76,77,78,79,80,81,82,83,84,85,86,87,88,89].

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
