# Peer review of "Analysis of Mining-Related Injuries in Chinese Coal Mines and Related Risk Factors: A Statistical Research Study Based on a Meta-Analysis"

_ijerph, 2022, doi:10.3390/ijerph192316249_

Round 1

Reviewer 1 Report

Although the paper analyzes a topic of interest, there are several weaknesses that have to be addressed before its publication.  They are discussed below.

The proposed analysis model seems too linear and too narrow not going deep enough in the analysis. Consequently, it does not adequately take into account the complexity of the operational environment (there are new hazards and risks related to the complexity of operational environment and organizational and human performance) as well as the influence of biases in decision making (motivational and cognitive) leading to accidents. Safety culture/management role such as presented in the model do not capture it well in this new context.

Thus, the scope of analysis is not sufficient. Such an approach often leads to putting the blame on frontline workers for deficiencies which are at the organizational level enabling and tolerating conditions creating unsafe workplaces (leading to the “drift to failure/accident”). The practice shows that these influences are “soft factors” that are hard to resolve, and may easily be hidden by apparent factors. Recent research works and return of experience show that the organizational performance and safety culture play a key role in creating conditions for accidents. The organizational performance also includes less studied motivational biases in decision-making process (mentioned above), and it is not considered either (cognitive biases are relatively well studied and understood).

Thus, the scope of the paper does not enable to capture the true image of the analyzed topic. Given the limited scope of the paper, the literature review is also too narrow, and it should be expanded to include aspects discussed above.

It is strongly recommended that the authors consult the following references while revising the paper.

Dekker S, Cilliers P, Hofmeyr, J.H., (2011), The complexity of failure: Implications of complexity theory for safety investigations. Safety Science. 49: 939-945

Kahneman, D. Thinking, Fast and Slow, Farrar, Straus and Giroux, New York: 2012

Komljenovic, D., Loiselle, G., Kumral, M., (2017), Organization: a new focus on mine safety improvement in a complex operational and business environment, International Journal of Mining Science and Technology, 27: 617-625

Leveson N.G., (2011a), Engineering a Safer World, Systems Thinking Applied to Safety. Cambridge MA: The MIT Press

Leveson N.G., (2011b), Applying system thinking to analyze and learn from events. Safety Science, 49:55-64

Montibeller, G. and Winterfeldt, D. Cognitive and Motivational Biases in Decision and Risk Analysis, Risk Anal. 2015: 35 (7): 1230-1251

Mosey, D. Looking beyond operator – Putting people in the mix, NEI Magazine, 2014; http://www.neimagazine.com/features/featurelooking-beyond-the-operator-4447549/
in collaboration with Ken Ellis, Managing Direction of World Association of Nuclear Power Operators (WANO)
http://www.neimagazine.com/features/featureputting-people-in-the-mix-4321534/

http://www.neimagazine.com/features/featureputting-people-in-the-mix-part-2-4322674/

Reviewer 2 Report

Manuscript includes literature-based statistic research. Its main objectives are statistical analysis of the characteristics of injuries to coal miners in China. The choice of topic is justified due to the number of employees in this sector, which exceeds three and a half million people in China, and the consequences of accidents, such as impact on workers' lives and family wellbeing.

The manuscript title describes its content well, and the abstract characterizes the problem, the research method and the results obtained in a systematic way.

The publication contains the correct selection and evaluation of the literature as well as its in-depth analysis. The research method is appropriate for the analyzed problem. The obtained results describe the criteria for injuries to coal miners in an accurate manner. They present main type of accidents, types of work, injured of the body parts, risk factors associated with casualties. The discussion includes not only the commentary on the data characterizing the quantitative results, but also recommendations for the improvement of safety.

The publication clearly defines its limitations and indicates the next steps of research, such as the need to confirm relationships between the study variables by validation in an actual production environment.

Detailed comments:

line 134: for consistency, add: “After presenting / describing the first part, there are others such as: .. "

Reviewer 3 Report

The investigation presents a coherent structure and design. It is well-founded, through a review of current and adequate literature. The introduction is structured and reasoned but, in my opinion, presents results that could be addressed in the discussion chapter, which is modest. See lines 75 to 92, where a detailed description of accident-related statistics is given, which could be, unless there is a better opinion, scrutinized in the discussion, enriching it.

In the discussion regarding the location of the injury, the authors focus attention on the availability and correct use of PPE. However, I recognize that, particularly in the mining sector, where the risk is high and in some cases considered extreme, the adoption of collective and organizational protection measures should be taken into account, considering the general principles of prevention. In this sense, authors should consider the discussion of collective protection protection measures, which are priorities, in relation to PPE.

Round 2

Reviewer 1 Report

The authors brought significant modification in the paper and adequately addressed the comments.